# Overhauser Dynamic Nuclear Polarization Enhanced Two-Dimensional Proton NMR Spectroscopy at Low Magnetic Fields

Timothy J. Keller, Thorsten Maly

5    Bridge12 Technologies Inc., 37 Loring Drive, Framingham, MA 01702, USA

*Correspondence to*: Thorsten Maly (tmaly@bridge12.com)

**Abstract.** The majority of low-field Overhauser Dynamic Nuclear Polarization (ODNP) experiments reported so far have been 1D NMR experiments to study molecular dynamics and in particular hydration dynamics. In this work, we demonstrate the application of ODNP-enhanced 2D J-resolved (JRES) spectroscopy to improve spectral resolution beyond the limit imposed

10    by the line broadening introduced by the paramagnetic polarizing agent. Using this approach, we are able to separate the overlapping multiplets of ethyl crotonate into a $2^{nd}$ dimension and clearly identify each chemical site individually. Crucial to these experiments is interleaved spectral referencing, a method introduced to compensate for temperature induced field drifts over the course of the NMR acquisition. This method does not require additional hardware such as a field-frequency lock, which is especially challenging when designing compact systems.

# 1    Introduction

In recent years, Dynamic Nuclear Polarization (DNP) has become a robust tool to boost the signal intensity of Nuclear Magnetic Resonance (NMR) experiments. The method has found widespread application in the areas of structural biology, imaging, and materials science. Besides increasing NMR sensitivity, DNP can be further used to obtain dynamic or spatial information about the system under study – information, which is often complementary to other methods (Ardenkjaer-Larsen, 2019, 2016; Corzilius, 2018; Griffin et al., 2019; Jaudzems et al., 2019; Kaminker, 2019; Liao et al., 2018; Maly et al., 2008; Plainchont et al., 2018; Rankin et al., 2019; Rosay et al., 2016).

Currently, the majority of DNP-NMR experiments are DNP-enhanced solid-state NMR (ssNMR) experiments often under Magic Angle Spinning (MAS) conditions. These experiments are typically performed at cryogenic temperatures (< 100 K) and commercial equipment is readily available (Rosay et al., 2016). In contrast to solution-state DNP experiments, DNP-enhanced ssNMR experiments require less demanding instrumentation since the sample can be directly irradiated by the microwave radiation without the use of a resonator. Due to the large cooling power of the cold nitrogen gas used to spin the samples for MAS experiments the sample temperature can be kept low. Only minimal microwave induced sample heating is observed, because of the low dielectric losses of a frozen aqueous sample $(\tan(\delta) < 0.01)$. (Nanni et al., 2011; Rosay et al., 2016).

For solution-state DNP experiments, the large dielectric losses of a liquid sample (e.g. water, see Figure 4 in (Neumann, 1985)) present a large challenge to high-field, solution-state DNP experiments. Particularly, in the case of aqueous samples, direct irradiating of the sample with microwave power results in excessive and often destructive sample heating. To avoid sample heating, a microwave resonator is required to separate the microwave induced electric fields, responsible for sample heating, from the microwave induced magnetic fields required to drive the DNP process (Poole, 1967).

Many microwave resonator types and structures are known from Electron Paramagnetic Resonance (EPR) experiments. However, since the overall dimensions of an EPR resonator scale with the wavelength of the required microwave radiation, resonators become physically very small at conventional NMR frequencies (Maly et al., 2008; Poole, 1967). For example, DNP at a proton Larmor frequency 400 MHz requires microwave radiation at 263 GHz, corresponding to a wavelength of < 1.4 mm. The resulting resonator geometries are difficult to fabricate and sample and resonator handling is very difficult.

In addition, an RF coil is required to detect the NMR signal which ideally should be located inside the microwave resonator, to assure a large NMR filling factor. If the RF coil is located outside the resonator, the gain due to DNP is easily offset by the reduced filling factor of the RF coil. However, Bennati et al. demonstrated that large solution-state DNP enhancement factors can be achieved at 95 GHz (Bennati and Orlando, 2019; Liu et al., 2017). Another promising avenue for high-field, solution-state DNP experiments are microfluidic structures in combination with strip line resonators and several research groups are active in this area (Denysenkov et al., 2017; Denysenkov and Prisner, 2019; K. Kratt et al., 2010; Webb, 2018)). Alternatively, overmoded photonic band gap structures can be used for solution-state DNP, however, RF filling factors

are typically low for these devices (Nevzorov et al., 2018). Direct polarization through DNP at high magnetic fields (14.1 T)
of large sample volumes (> 100 μl) without using a resonator is possible in unpolar solvents (Dubroca et al., 2019). However, the microwave induced sample heating limits this method to studying samples in solvents that have only small ohmic losses (e.g. small molecules in organic solvents), but even then chemical shift referencing will be challenging.

Over the past two decades, these technical challenges have led to the development of different strategies to avoid microwave induced sample heating such as the dissolution DNP (dDNP) experiment pioneered by Adenkjaer et al. (Ardenkjær-Larsen et al., 2003) in which the sample is polarized at low magnetic fields prior to melting it and quickly transferring it to a high-field spectrometer for NMR acquisition. The method can be used for analytical chemistry (Chen et al., 2013; Chen and Hilty, 2015) but is typically used to generate polarized solutions for Magnetic Resonance Imaging (MRI) experiments (Ardenkjaer-Larsen, 2019, 2016). Although dDNP experiments are one-shot experiments, which cannot be repeated using the same sample, more dimensional experiments can be performed using rapid NMR methods (Frydman and Blazina, 2007). A variation of the dDNP experiment is the temperature jump experiment, in which the sample is polarized prior to melting the sample *in-situ* using a powerful laser. Once the liquid-state NMR spectrum is recorded, the laser is switched off, the sample freezes and can be repolarized (Joo et al., 2009, 2006; Sharma et al., 2015). Alternatively, the liquid sample can be polarized at low magnetic fields (e.g. 0.35 T) and either rapidly shuttled into a high-field NMR spectrometer (e.g. 14.2 T) (Krahn et al., 2010; Reese, 2008; Reese et al., 2009) or transferred to the high field magnet using a pump (Dorn et al., 1989, 1988; Liu et al., 2019). However, the polarization decreases during the transfer time, compromising the sensitivity gain achieved by the ODNP process. In addition, polarizing at a low magnetic field and detecting the NMR signal at a higher magnetic fields results in a "Boltzmann factor penalty" (Fedotov et al., 2020). For example, polarizing the sample at 0.35 T (15 MHz proton frequency) and detecting in a benchtop NMR system operation at 1.88 T (80 MHz proton frequency) results in a penalty factor of > 5 and acquiring the NMR signal at the same field at which it was polarized will always result in the largest overall sensitivity gain compared to methods that polarize at a low field and detect the signal at high fields.

Here, we employ Overhauser Dynamic Nuclear Polarization (ODNP) spectroscopy to enhance the signal intensity in a low-field NMR experiment. The Overhauser effect causes a polarization transfer from an electron spin to a nuclear spin when the electron spin transition is irradiated. The amount of polarization transferred ($\varepsilon$) is given by the classic equation (Hausser and Stehlik, 1968):

$$\varepsilon = 1 - E = \xi f s \left| \frac{\gamma_S}{\gamma_I} \right| \qquad (1)$$

Where $E$ is the enhancement, $\xi$ is the coupling factor, $f$ is the leakage factor, $s$ is the electron saturation factor, $\gamma_S$ is the electron gyromagnetic ratio, and $\gamma_I$ is the nuclear gyromagnetic ratio. The coupling factor can vary from -1 in the case of pure scalar coupling to +0.5 in the case of pure dipolar coupling. For protons and nitroxides in solution, the interaction is almost entirely dipolar which yields a maximum possible enhancement of -330. Hausser et al. showed already in their early work that the Overhauser DNP effect is most efficient at low magnetic field strengths (Hausser and Stehlik, 1968). The

coupling factor dramatically decreases at higher magnetic fields leading to lower enhancement factors with increased magnetic field strengths.

Solution-state ODNP spectroscopy at low magnetic field strengths is no new method and was extensively used in the 1960s to the 1970s (Hausser and Stehlik, 1968; Mueller-Warmuth et al., 1983). However, the method almost completely vanished with the push of magnetic resonance methods to ever higher magnetic fields but was resurrected in the early 2000s because of its potential to determine local hydration dynamics on surfaces (Armstrong and Han, 2007). Today, it is an active field of research and the theory, instrumentation and application of ODNP spectroscopy is constantly developing (Armstrong and Han, 2009; Doll et al., 2012; Franck, 2020; Franck et al., 2013; Franck and Han, 2019; Han et al., 2008; Keller et al., 2020). While the observed enhancements can be large at low fields, a major drawback of low-field NMR spectroscopy, with or without the addition of ODNP, is the decreased resolution. However, the techniques remains attractive for applications in which some of the resolution can be sacrificed at the benefit of greatly simplified instrumentation.

In an earlier publication we demonstrated that high-resolution, solution-state ODNP-enhanced NMR spectra can be recorded at low magnetic fields (0.35 T, 14 MHz [1]H Larmor frequency) (Keller et al., 2020). Performing ODNP spectroscopy at this field has the advantage that instrumentation is readily available from X-band (9.5 GHz) EPR spectroscopy. In addition, enhancement factors are typically large, because the ODNP effect scales favourable with decreasing magnetic fields (Hausser and Stehlik, 1968; Kucuk et al., 2015; Sezer, 2014). Here, we present for the first time, ODNP-enhanced two-dimensional (2D) high-resolution proton NMR spectra of small molecules recorded at a magnetic field strength of 0.35 T using a highly homogenous permanent magnet. While [19]F 2D ODNP-enhanced spectroscopy has have been reported previously, the small chemical shift dispersion of protons make these experiments especially challenging (George and Chandrakumar, 2014). At a higher field of 1.2 T, ODNP experiments with 2D heteronuclear correlation (HETCOR) have been performed (Dey et al., 2017).

Experiments in this work were performed on a compact, home-built DNP/NMR system using a permanent magnet. Steps have been taken to compensate for temperature induced magnetic field drift of the permanent magnet, which makes these experiments difficult. To mitigate these adverse effects and to obtain high-resolution spectra, we introduce a novel acquisition scheme and processing workflow.

## 2    Material and Methods

### 2.1    Chemicals

4-Oxo-2,2,6,6-tetramethyl-1-piperidinyloxy (TEMPONE), 4-hydroxy-2,2,6,6-tetramethylpiperidine 1-oxyl (TEMPOL), ethyl crotonate, and ethanol were purchased from Sigma-Aldrich. All chemicals were used without further purification.

## 2.2 Sample Preparation

For 10 mM TEMPONE in ethyl crotonate, 5 mm sample height was loaded into 0.98 mm ID, 1.00 mm OD quartz capillary (Hampton Research, HR6-146). For 10 mM TEMPOL in ethanol, 5 mm sample height was loaded into 0.60 mm ID, 0.84 mm OD quartz capillary (VitroCom, CV6084-Q-100).

## 2.3 ODNP Spectrometer

All ODNP experiments were performed in a home-built spectrometer, which requires four principal components: 1) a high-power microwave source, 2) a microwave resonator with integrated NMR coil, 3) an NMR spectrometer, and 4) a magnet. We used a home-built microwave source with a maximum output power of 10 W, which operates over a frequency range of 8 to 12 GHz. A home-built, dielectric resonator operating in the $TE_{011}$ mode at a frequency of 9.75 GHz with integrated saddle coil was used in all experiments, with a loaded quality factor Q of 6900. The Q dropped to > 4000 when a (aqueous) sample was inserted. The resonator is coupled to the waveguide by means of a circular iris. To optimize coupling to the resonator, the reflected power from the cavity was monitored and minimized by adjusting an iris screw. We estimated about 1.5 dB loss from the output of the microwave source to the sample position. Power levels given throughout this article correspond to the estimated microwave power levels at the position of the sample. The microwave source constantly monitors the forward (Tx) and reflected (Rx) microwave power. During operation the resonator can heat up, resulting in a shift of the resonator frequency and the microwave frequency is automatically adjusted to minimize the reflected microwave power (lock mode).

NMR experiments were performed using a Kea2 spectrometer (Magritek), with an external RF amplifier (MiniCircuits, model LZY-22+). At the nominal output power of the RF amplifier of > 30 W, the observed NMR pulse length for a 90°-pulse was about 5 μs.

All experiments are performed using a permanent dipole magnet (SABER Enterprises, LLC., North Andover, MA USA). The nominal field strength of the magnet is 0.35 T, with a native homogeneity at the position of the sample of < 10 ppm. The magnet is equipped with a $B_0$ sweep coil to make small adjustments (+/- 15 mT) to the field strength. To perform high-resolution NMR spectroscopy, a set of electric shims are used. Shim coils were fabricated from printed circuit boards mounted to the magnet pole faces and included the zonal correction coils Z1 and Z2 and the tesseral correction coils X and Y. The physical dimensions of the coils were determined following the procedure outlined by Anderson (Anderson, 1961). Two triple channel power supplies (HP Model 6623A) were used to drive the sweep and shim coils. We observed a native linewidth of a (tap) water sample without energizing the shim coils of 110 Hz (8 ppm). With shim-coils energized and optimized, we observed a linewidth of < 2.3 Hz (0.16 ppm) for a water sample with 200 μM TEMPOL (see **Figure S2**). Typically, the current for the electrical shims was optimized before each experiment. To compensate for ambient temperature fluctuations, the magnet was placed inside a small lab incubator. The temperature of the incubator was set to 32 ºC for all experiments. A picture of the

experimental setup is shown in **Figure S1**. To cool the sample, dry air was continuously flowed through the resonator at a rate of 2 L/min.

## 2.4 ODNP Experiments:

The microwave power for all ODNP experiments was set to 35 dBm (3.2 W). The native resonance frequency of the ODNP resonator was found to be 9.75 GHz, corresponding to a proton NMR frequency of 14.7945 MHz. Continuous wave (cw) microwave irradiation was used for all experiments. The sweep coils of the magnet were set to maximize the ODNP enhancement.

        1D proton ODNP spectra were acquired using an in-house developed pulse program which allows for each phase
cycle/average to be stored individually along a 2nd dimension. 1D proton ODNP spectra were acquired using a repetition time of 2 s and a total of 128 transients using a 4-step phase cycle. Four dummy scans were performed before each acquisition to establish thermal equilibrium. The FID contained 8192 points with a dwell time of 200 μs. A 90º-pulse length of 5 μs was used.

        J-Resolved (JRES) experiments were acquired using an in-house developed pulse program to save each phase cycle
separately. A 1d proton reference spectrum was acquired after each phase cycle of the JRES experiment was completed (interleaved spectral referencing). Prior to each spectrum recorded in the $t_1$ dimension, 2 dummy scans were used to equilibrate the magnetization during the experiment. The repetition time was set to 2.5 s and a 4-step phase cycle was used to eliminate artifacts (Berger and Braun, 2004). The spin echo was acquired with 4096 points and a dwell time of 200 μs. The indirect dimension was acquired with 128 points and a delay increment of 8 ms (initial inter-pulse delay of 3 ms). The 90°- and 180°-
pulse lengths were set to 10 μs.

## 2.5 Data Processing

        All spectra were processed using DNPLab, an open-source python package for processing DNP data (https://github.com/DNPLab/DNPLab). The package is developed in collaboration between Bridge12 Technologies, Songi Han's lab at UCSB, and John Franck's lab at Syracuse University. DNPLab is able to import various spectrometer formats
(e.g. Topspin, (Open) VnmrJ, Prospa, Tecmag etc.) and converts the data into a versatile python class for manipulating N-dimensional data arrays. Standard processing functions for NMR data can be easily applied along any specified dimension. In addition, DNPLab is flexible enough to allow experienced python users to perform custom processing if desired. A complete description of DNPLab will be subject to a forthcoming publication.

        To process 1D proton spectra, a window function was applied, prior to Fourier transformation of the FID. Averages
were aligned using a FT cross correlation method (Vu and Laukens, 2013) and summed together to generate the final spectrum.

The JRES data was corrected for field drift. A detailed description is given in Section 3.1 Interleaved Spectral Referencing for Magnetic Field Drift Correction. Typically, a Lorentz-Gauss transformation was applied along the direct and indirect dimension. Prior to Fourier transformation, the data was zero filled to twice the original length in both dimensions. After Fourier transformation, a shearing transformation was applied and the JRES spectrum was symmetrized using the geometric average (Ernst et al., 1991). The skyline projection was acquired by taking the maximum signal intensity along the indirect dimension. All ODNP enhanced spectra, which have negative enhancements, are phased positively. No internal referencing standard was used. NMR spectra of Ethyl crotonate were referenced according to values of the protons of the methyl group (at 1.28 ppm) as given by Berger et al. (Berger and Braun, 2004). NMR spectra of ethanol and water were referenced according to values given in Fulmer et al. (Fulmer et al., 2010).

All experimental raw data is available in GitHub (DOI: 10.5281/zenodo.4479048) (Keller and Maly, 2021).

## 3  3 Results and Discussion

To demonstrate the feasibility of low-field ODNP-enhanced 2D NMR spectroscopy we choose two small molecules: ethanol and ethyl crotonate. While the NMR spectrum of ethanol is relatively simple (see SI for details), our discussion will focus on the use of ethyl crotonate. The NMR spectrum of ethyl crotonate is well-understood. The molecule has a variety of different proton sites with a large dispersion of proton chemical shifts and J-couplings, resulting in a "crowded" low-field NMR spectrum. The molecular structure for ethyl crotonate is given in **Figure 1**.

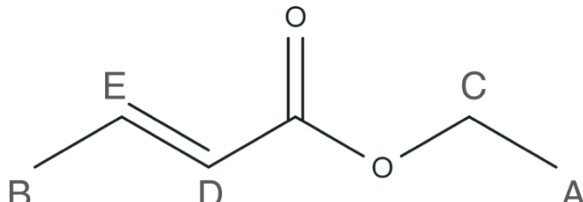

**Figure 1:** Molecular structure of ethyl crotonate and proton labels used throughout this work.

For the ethyl crotonate sample under study, we typically observe a signal enhancement of about -30, which is on the lower side. We attribute this to two factors: 1) the viscosity, and 2) the sample temperature. In our studies we used a sample of 10 mM TEMPONE in neat ethyl crotonate. This sample has a higher viscosity than water, which will lead to lower overall enhancements (Hausser and Stehlik, 1968). Another factor, that strongly influences the achieved enhancement is the sample temperature. Higher sample temperatures will lead to higher enhancements. However, due to the large resonator Q factor and the good separation of the microwave induced electric and magnetic fields, sample heating is strongly minimized (Keller et

al., 2020). However, we would like to note that even a moderate enhancement of about -30 will lead to a time saving of factor 900.

### 3.1 Interleaved Spectral Referencing for Magnetic Field Drift Correction

In general, magnetic field drifts cause line broadening and/or artifacts in NMR spectra and numerous methods have been implemented for solution and solid-state NMR experiments to compensate for these adverse effects. These hardware-based methods typically use a lock signal from a reference sample or nucleus (e.g. deuterium) and either the field or the NMR transmitter frequency is adjusted accordingly (Baker and Burd, 1957; Maly et al., 2006; Markiewicz, 2002; Paulson and Zilm, 2005). In contrast, software-based methods can be also used to account for magnetic field drifts (Ha et al., 2014; Najbauer and Andreas, 2019).

Using a software-based method to correct for the field drift has the advantage that no additional hardware is required. We used a compact, single channel NMR spectrometer with no lock channel. This type of hardware is commonly found in NMR applications for process monitoring or well-logging applications, which often have strict space restrictions and adding additional hardware is challenging or simply not possible (Ha et al., 2014; Mandal et al., 2014; Song and Kausik, 2019).

Correcting for magnetic field drift can be particularly challenging for permanent magnets utilizing rare earth magnets which have a large temperature coefficient causing the field to be susceptible to small fluctuations in room temperature. In such magnets it is common to stabilize the temperature to ≤100 mK (Windt et al., 2011) in addition to using a field frequency lock (Blümich, 2016; Danieli et al., 2010).

The temperature drift will cause a slow drift over the course of minutes or hours. Higher frequency fluctuations caused by magnetic coupling to the environment will also lead to line broadening and/or artifacts. Ripple from shim power supplies can cause line broadening and/or sidebands at the mains frequency. In addition, PID temperature controls which typically use pulse width modulation (PWM) to control heaters can cause field shifts and/or oscillations.

In our experiments, we employ two strategies to minimize artifacts from field drift. Any high frequency fluctuations were suppressed by disabling the heater during NMR acquisition. In this case, the temperature will no longer be regulated, and the field will start to drift. To correct this, we acquired a reference 1D proton NMR reference spectrum after each complete phase cycle of the JRES experiment (Interleafed Spectral Referencing, see **Figure 2 (a)**). Since we only record a reference spectrum after each phase cycle is completed, the overall measurement time is only slightly increased, adding only little overhead. For the JRES experiments, we used a 4-step phase cycle, therefore, interleaved spectral referencing only adds 20 % overhead. If the drift is small, fewer reference spectra can be recorded, further reducing the amount of overhead.

Over the period of acquiring the 2D JRES spectrum we typically observe an overall field drift of approximately 60 ppm over the course of ~30 minutes. The change in chemical shift was calculated relative to the first spectrum and the change in chemical shift for each JRES step was then fit to a $4^{th}$ order polynomial **Figure 2 (b)**. We found a $4^{th}$ order polynomial to be sufficient resulting in a residual < +/- 0.05 ppm (see **Figure 2 (c))**, significantly smaller compared to the observed proton linewidth of ~3 Hz (0.2 ppm). Each transient of the JRES experiment, including different phase cycle steps, were stored

separately so that a field drift correction could be applied before the phase cycle. In addition to chemical shift, the 1D proton reference spectra were also used to correct the spectral phase of each step in the JRES experiment (a detailed description is given in the SI). In addition to the field drift, we also observed a phase drift over the course of the experiment, which we attribute to two different effects: 1) the acquisition delay (spectrometer deadtime) will result in a linear phase roll in the frequency domain as the field drifts, and 2) instrumental and environmental instabilities (e.g. ambient temperature fluctuations in the room). All these effects can be corrected for by interleaved spectral referencing. All processing was performed using the python package DNPLab.

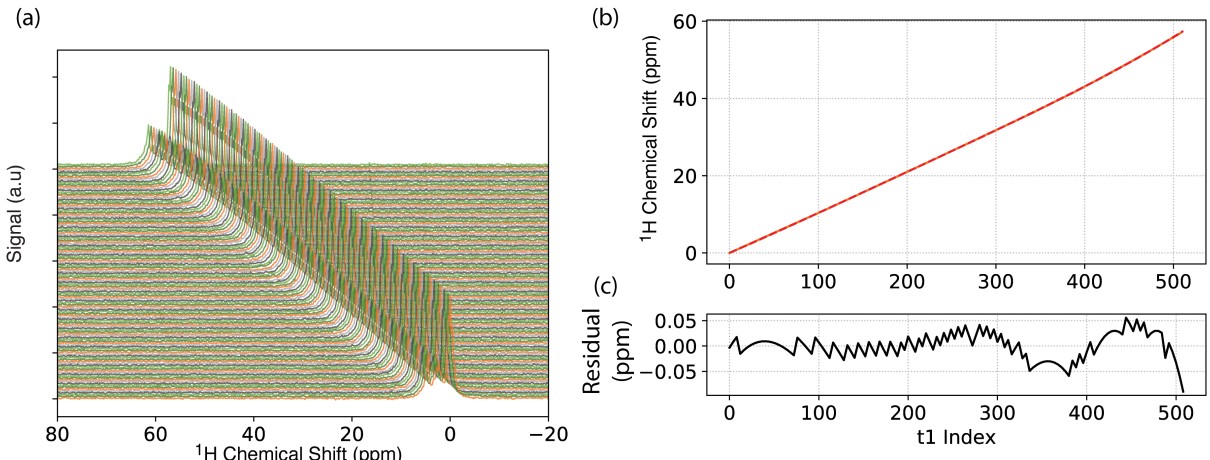

**Figure 2:** (a) 1D ODNP-enhanced ethyl crotonate reference spectra with 10 mM TEMPONE acquired after each completed phase cycle step of JRES experiment. Spectra are vertically offset for clarity. (b) Chemical shift of the peak with the maximum intensity and fit (4th order polynomial) for experimental drift correction. (c) Residual of the fit.

Interleaved spectra referencing is a general concept, which is not restricted to any particular NMR sequence. The frequency of the acquisition of the reference spectrum depends on the scale of the observed field drift, only adding very little overhead to the entire acquisition. Acquiring a separate 1D spectrum has the additional benefit that the signal intensity will be always constant. In a 2D experiment transients with longer evolution times often don't have a sufficiently large enough signal-to-noise ratio. In the two presented case for ethyl-crotonate and ethanol we only observed a monotonic drift of the field. However, as long as a function can be found that adequately models the field drift it will be possible to correct for the drift even for non-monotonic drifts (e.g. oscillations) or sudden field jumps.

### 3.2 Resolution Enhancement

In **Figure 3** (top), a 1D proton NMR spectrum of ethyl crotonate is shown together with the resonance assignment corresponding to the protons of the molecule. All resonances can be identified and assigned. We applied a 1 Hz Lorentzian apodization window resulting in an average native linewidth of 7.3 Hz (see the SI for details). As demonstrated in an earlier publication, the presence of the polarizing agent (TEMPONE) does not prevent us from resolving the J-couplings in ethyl

crotonate (Keller et al., 2020). However, the polarizing agent does introduce significant line broadening at a concentration of 10 mM and the linewidth is limited by the polarizing agent, not magnetic field homogeneity. The linewidth limited by the magnetic field is 2.3 Hz (0.16 ppm) as shown for a sample of 200 µM TEMPOL in water (see SI, Figure S2).

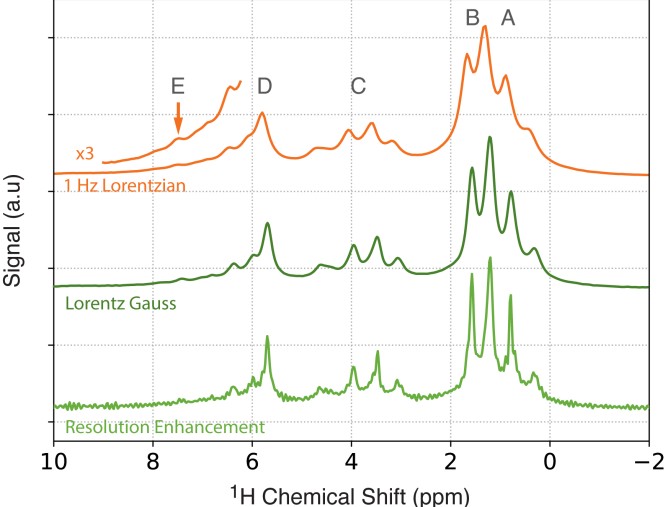

**Figure 3:** 1D ODNP-enhanced NMR spectra (128 averages) of ethyl crotonate with 10 mM TEMPONE using different apodization functions. Top: 1 Hz Lorentzian line broadening. Middle: Lorentz-Gauss transformation using 4 Hz line broadening for both Gaussian and Lorentzian linewidths. Bottom: Resolution enhancement after Trafficante et al. using $T_2^*$ of 0.2 s.

The spectral resolution can be further improved using common resolution enhancement techniques such as Lorentz-Gauss transformation or other windowing functions (Ernst et al., 1991; Trafficante and Nemeth, 1987; Trafficante and Rajabzadeh, 2000). Typically, these methods are known to decrease the signal-to-noise-ratio, however, because of the dramatic improvement in sensitivity when using ODNP, some loss in spectral resolution for low-field NMR experiments can be regained. This is especially attractive, since the observed ODNP enhancements are larger at lower fields. The results for two commonly used window functions for resolution enhancement are shown in **Figure 3** (middle and bottom). By using the Lorentz-Gauss transformation, the overall spectral linewidth can be reduced from 7.3 Hz, as observed in the NMR spectrum process using a 1 Hz Lorentzian line broadening, to 5.6 Hz (**Figure 3**, middle). Further reduction in linewidth can be achieved using the method reported by Trafficante et al., however, at the expense of some increased artifacts (**Figure 3**, bottom). Using this window function a linewidth of 4.4 Hz was observed. Both, the Lorentz-Gauss transformation as well as the resolution-enhancement window function improve the observed linewidth. However, for the 2D JRES experiments we used the Lorentz-Gauss transformation, since it introduces fewer artifacts. For a detailed discussion about the choice of the apodization window the reader is referred to the discussion in the Supporting Information.

 **3.3    ODNP-Enhanced ¹H JRES Experiments on Ethyl Crotonate**

At low magnetic fields, overlapping J-couplings can lead to crowded, unresolved spectra. One method to simplify complicated spectra is JRES spectroscopy which separates the J-coupling along a 2nd dimension (Aue et al., 1976). This technique is commonly used in metabolomics (Ludwig and Viant, 2010). The JRES experiment is the simplest experiment of methods belonging to a group of experiments commonly referred to as pure-shift experiments (Aguilar et al., 2010; Zangger, 270 2015). We used the JRES method, the simplest implementation of pure-shift spectroscopy, because it does not require the use of pulsed field gradients.

JRES experiments were performed on samples of two small molecules, ethanol and ethyl-crotonate. Due to the limited features observed in the JRES experiment, the data for ethanol are shown in the SI. The ODNP-enhanced JRES spectrum of ethyl crotonate with 10 mM TEMPONE is shown in **Figure 4 (a)**. For each chemical shift corresponding to a specific proton 275 site a 1D slice is given showing the pattern caused by the J-coupling for each proton site (**Figure** 4 **(b),** labels correspond to protons as shown in **Figure 1**). The protons of the two methyl groups (protons A and B) which are overlapping in the 1D NMR spectrum (**Figure 3**), can be clearly resolved in the 2D JRES spectrum. The same is true for the methylene quartet (proton C) which has some overlap with proton D in the 1D spectrum. The measured J-couplings are 7±1 Hz for J(H-A, H-C), 7±1 Hz for J(H-B, H-E), and 16±1 Hz for J(H-D, H-E). The observed values are in excellent agreement to within 1 Hz with values 280 published in the literature (Berger and Braun, 2004).

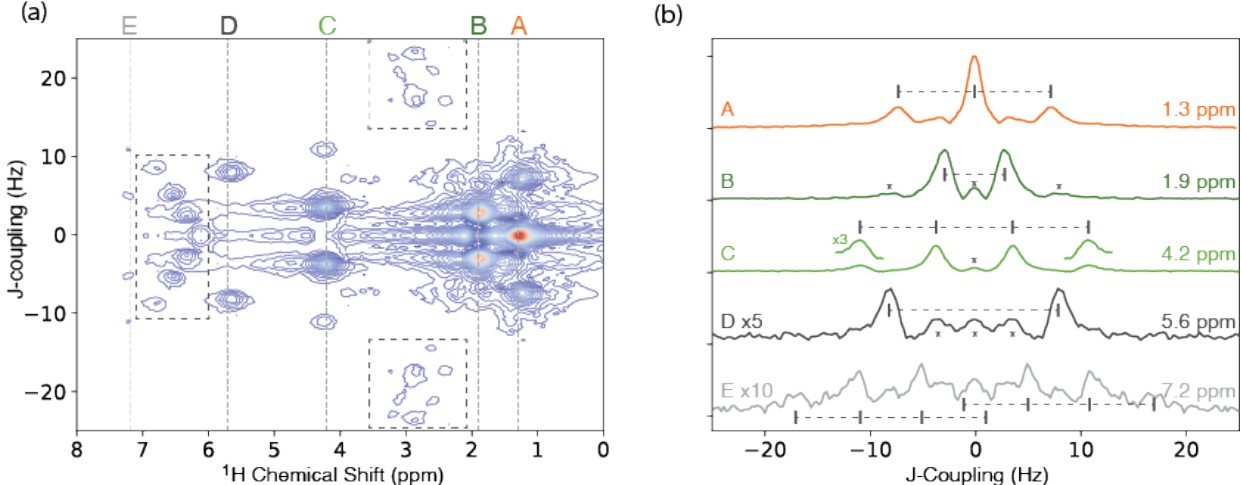

**Figure 4:** (a) ODNP-enhanced ¹H JRES spectrum of neat ethyl crotonate with 10 mM TEMPONE. The dashed boxes indicate areas showing strong coupling effects. (b) Slices of JRES spectrum shown in (a) for each proton in ethyl crotonate. The expected multiplet pattern is indicated by vertical lines with corresponding tick marks. Spectra are offset for clarity. Features marked with an asterisk are artifacts of the JRES experiment.

To obtain a 1D spectrum, in which the J-coupling is removed (pure-shift spectrum), a skyline projection of the JRES spectrum along the J-coupling dimension can be calculated (see **Figure 5**, orange). While this simplifies the spectrum, the

peak intensities are not quantitative anymore. For a quantitative analysis, the peak intensities have to be integrated along the indirect dimension, which will allow for accurate measurements of the enhancements for each chemical site (see SI). In the skyline projection, the peaks corresponding to the methyl groups (protons A and B) are clearly separated, compared to a regular 1D proton spectrum (**Figure 5**, green). In addition, the quartet located at 4.2 ppm in the regular 1D spectrum collapses to a single, well resolved line in the skyline projection located at a chemical shift of 4.2 ppm.

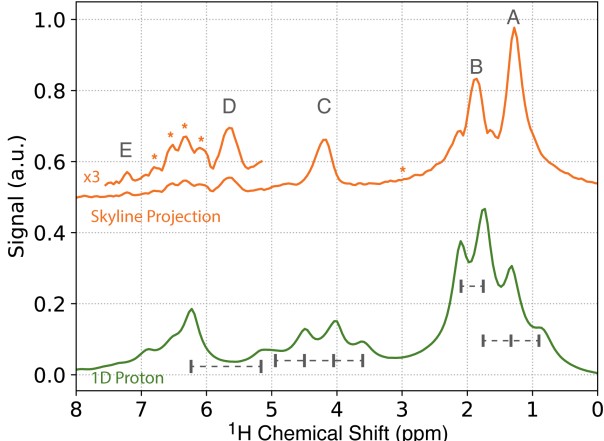

**Figure 5:** NMR spectra of ethyl crotonate. Top: Skyline projection of 2D JRES experiment. The locations of strong-coupling artifacts are indicated by asterisks. Bottom: 1D 1H NMR spectrum.

It is a well-known fact that JRES spectra suffer from artifacts resulting from strongly coupled spin systems. These artifacts cannot be removed by phase cycling or pulsed field gradients, since these artifacts originate from the same coherence transfer path as the desired signal. These artifacts lead to additional lines in the skyline projection which are not at the position of the chemical shift (Ludwig and Viant, 2010; Thrippleton et al., 2005). For ethyl crotonate, we observe strong-coupling artifacts for protons A and C at a chemical shift of about 3 ppm and a J-coupling of +/- 20 Hz in the ODNP-enhanced JRES spectrum. Furthermore, peaks due to strong coupling effects can be observed in the region of 6 to 7 ppm. These peaks are attributed to the coupling between protons D and E and result in a complicated multiplet structure (see dashed boxes in **Figure 4 (a)**). These unwanted peaks also show up in the skyline projection shown in **Figure 5** (orange, marked with an asterisk). However, they are easy to distinguish in the full 2D spectrum and since the skyline projection is calculated from the 2D spectrum, it is relatively straight forward to separate wanted from unwanted peaks.

## 4    Future Directions

Currently, the majority of low-field, ODNP-enhanced NMR experiments are 1D experiments to study hydration dynamics. For these measurements a decent homogeneity of the magnetic field is sufficient since only a single peak (water) is

observed. However, improving the homogeneity of the magnetic field will not only lead to higher sensitivity due to improved lineshapes, but it also opens the possibilities of performing high-resolution ODNP-enhanced NMR spectroscopy. The use of

305 active shims can improve spectral resolution to the linewidth limited by the paramagnetic broadening effects of the polarizing agent. Any further improvement in resolution must be gained by other techniques such as resolution enhancement (data processing) and/or multi-dimensional NMR experiments and the possible applications are plentiful.

As new applications, high-resolution ODNP-enhanced JRES spectroscopy enables site-specific, chemical shift resolved solvent dynamics measurements. For example, ODNP measurements on toluene show different saturation behaviour

for the methyl and aromatic protons in toluene suggesting different solvent dynamics for the different proton sites (Enkin et al., 2014; Keller et al., 2020). Since the peak separation is only about 4 ppm, highly homogenous magnetic fields are required to study the peaks individually since overlapping peaks will make data analysis more difficult. These types of experiments can greatly benefit from JRES spectroscopy. In contrast to correlation type experiments such as COSY, the individual proton sites are resolved in the indirect dimension in the JRES experiment. This will allow detecting the microwave power saturation

behaviour of an individual proton site. A correlation peak observed in for example a COSY experiment on the other hand will show the superposition of the saturation behaviour of two sites, further complicating the data analysis.

Another recent example is the study of water diffusion in polymer membranes such as Nafion. Water located inside the membrane (channel water) has a higher chemical shift of about 5.5 ppm, while residual water has a chemical shift of 4.7 ppm (Kim et al., 2016; Überrück et al., 2018). To study the diffusion behaviour using ODNP spectroscopy high-resolution

techniques are required in combination with multidimensional NMR methods. 2D ODNP-enhanced JRES experiments are particularly helpful because it allows studying complex systems by bringing site-specific resolution.

A common practice in low-field NMR spectroscopy is to record $T_1/T_2$ relaxation maps to identify different species in complex mixtures based on their relaxation behaviour (Colnago et al., 2021; Song et al., 2021). Other methods just rely on recording either $T_1$ or $T_2$ relaxation times and are successfully used in cancer diagnosis (Castro et al., 2014; Issadore et al.,

2011; Min et al., 2012). These point-of-care systems can potentially benefit from ODNP-enhanced, multi-dimensional, ODNP-enhanced low-field NMR spectroscopy, by not only studying the relaxation behaviour of a sample, but in addition, acquiring spectral information.

Another field that will benefit from the presented technique is NMR spectroscopy used in reaction monitoring (Dalitz et al., 2012; Plainchont et al., 2018). High-resolution ODNP-enhanced 2D NMR spectroscopy opens the possibility of studying

more complex mixtures. Portable low-field ODNP systems have been reported in the literature (Ebert et al., 2012; Münnemann et al., 2008) but so far these systems only achieve low to moderate spectral resolution. Typically, compact, portable systems are used in these applications that do not use a superconducting magnet. These systems are based on small electromagnets or permanent magnets, which can greatly benefit not only from ODNP spectroscopy, but also from Interleafed Spectral Referencing.

From here the possibilities of improvements are countless. Implementation of JRES methods that do not result in artifacts due to strong spin coupling effects are straight forward (Thrippleton et al., 2005). Furthermore, integration of pulse

field gradient coils into the magnet system will accelerate acquisition of spectral acquisition of multidimensional experiments. In addition, with strong enough gradient pulses ultrafast 2D methods introduced by Frydman et al. are possible (Gouilleux et al., 2018). Many of these concepts are currently implemented in our lab.

**5    Conclusion**

In this work, we demonstrate the application of ODNP-enhanced 2D JRES spectroscopy to improve spectral resolution beyond the limit given by the line broadening introduced by the paramagnetic polarizing agent. As of proof-of-concept we use the simplest implementation of the 2D JRES experiment and achieve full spectral resolution for small molecules at low magnetic fields.

We show that multi-dimensional NMR experiments can be applied to increase the resolution in low-field ODNP experiments. Using ODNP-enhanced 2D JRES spectroscopy, we are able to separate the overlapping multiplets of ethyl crotonate into a $2^{nd}$ dimension and clearly identify each chemical cite. While in most circumstances, the emphasis of (O)DNP-enhanced spectroscopy is on improvement in sensitivity, it should be noted that the improved sensitivity can allow more aggressive apodization of data to increase spectral resolution.

Crucial to these experiments is the interleaved spectral referencing to compensate for temperature induced field drifts over the course of the JRES experiment. This method does not require additional hardware such as a field-frequency lock, which is especially challenging when designing compact systems.

**6    Code availability**

All experimental data was processed using DNPLab, an open-source python package to process NMR, EPR and DNP
data. DNPLab is available for free download at https://github.com/DNPLab/DNPLab.

**7    Data availability**

Experimental     raw     data     is     publicly     available     on     GitHub     at https://github.com/Bridge12Technologies/2D_ODNP_Spectroscopy_DataRep.

DOI: 10.5281/zenodo.4479048

**8    Author Contribution**

TM designed the research and designed the experiments in collaboration with TK. All experiments were carried out and analyzed by TK.

## 9 Competing Interests

Authors TM and TK are employees of Bridge12 Technologies, Inc. TM is a co-founder of Bridge12 Technologies, Inc.

## 10 Acknowledgements

We thank Dennis Gautreau, Walter Hrynyk, Charan Gujjala, and Rohit Arora for the mechanical design, and Jagadishwar Sirigiri, Alexander Laut, John Franck (Syracuse University), Thomas Casey and Songi Han (University of California, Santa Barbara) for many stimulating discussions. This work was supported by a Small Business Innovation Research (SBIR) grants from the National Institute of General Medical Science (NIGMS) of the National Institutes of Health (NIH) grant GM112391 and GM128542.

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
