# Peer review of "Overhauser Dynamic Nuclear Polarization Enhanced Two-Dimensional Proton NMR Spectroscopy at Low Magnetic Fields"

_Magnetic Resonance, 2021_

## Author Response (AR1)

**Reviewer 1**

The authors present an important topic for Magnetic Resonance, the development of a new method in the area of 2D NMR-DNP to study liquid samples. They demonstrate clearly the newly developed approach for compact low field spectrometers. They solve a systematic problem, significant field drift, common with permanent magnets used in NMR spectrometers. Also, it is appreciated and important for the NMR community that their code is easily accessible. Finally, I recommend that the submitted paper be published with minor corrections. See below.

Possible corrections:
1. line 152: do you mean Tecmag, not Techmag, correct?
   **Response:** Yes, this is correct, we were referring to Tecmag. This mistake will be corrected in the revised version of the manuscript.

2. In Figure 4a, shouldn't the x-axis be labeled "J-coupling (Hz)"?
   **Response:** This mistake will be corrected in the revised version of the manuscript.

Suggested comments to address:
1. It seems from the technical details that 2D acquisitions presented took about 30 mins. What are the limitations of the proposed method? Could it be applied to non-monotonic drift? Could the method be applied for much longer acquisition times particularly when the samples are not neat compounds?
   **Response:** In the two examples presented here the field drift can be modeled using a 4$^{th}$ order polynomial (we will add the coefficients to the SI in the revised version of the manuscript). In these experiments, the field drift was always monotonic so we can only speculate about the performance of our field correction method. We do believe, as long as a function can be found that adequately models the field drift it will be possible to correct for the drift even for non-monotonic drifts (e.g. oscillations). In the case of sudden field jumps, it should still be possible to correct for such disturbances. This should allow for measurements that would require much longer acquisition times.
2. What applications would benefit most from such instrument/method? Compared to more traditional compact NMR spectrometers?
   **Response:** In recent years many more vendors offer bench-top NMR spectrometers. In our system, we demonstrate, as a proof-of-concept, that high-resolution NMR spectroscopy can be combined with ODNP. While we can foresee/speculate many different applications, one particular application, the measurement of hydration dynamics stands out as the method that would most benefit from the described instrument. Currently, ODNP measurements for hydration dynamics are performed by adding DNP capabilities to an EPR spectrometer. However, the homogeneity of the electromagnet is often not ideal, hence the off-signal (the signal recorded without microwave radiation) is often weak and broadened due to field drift/instabilities. While an EPR magnet can be equipped with active shims, a system based on a compact

permanent magnet will have a much smaller footprint compared to a floor-model EPR spectrometer. In addition, there are several new diagnostic techniques that rely on low-field NMR measurements (e.g. methods developed by T2 Biosystems, WaveGuide Corp., Synex Medical e.g.). These methods often rely on measurements of relaxation parameters because of the inhomogeneous magnetic field. These methods can potentially benefit from ODNP too. We will add this discussion to the revised version of the manuscript.

**Reviewer 2**

This manuscript demonstrates two-dimensional NMR with quite decent resolution at a proton frequency of about 14 MHz, where Overhauser DNP is more easily applicable than at higher field. The authors also show that the "pure-shift" JRES experiment can be quite useful even in the strong-coupling regime, where pure-shift spectra cannot be obtained. Further, they present a simple - though not necessarily optimal - technique for dealing with magnetic field and phase drifts. Altogether, this manuscript demonstrates sufficient progress for a publication. However, I find several arguments lacking, as detailed below. This particularly regards the application potential of the technique. If sensitivity enhancement is the main argument, then the presented data is unconvincing. This technique will find a niche if and only if it can be applied to problems where the transfer of polarization from electron to nuclear spins provides information rather than "only" sensitivity gain. The authors should respond to the following comments before I can make a final recommendation on publication.

**Intro:** We thank the reviewer for reading our manuscript in great detail. We believe that addressing the raised issues will improve the quality of the manuscript. Please find our detailed response below.

1. (l. 49-50): It is not clear to me why the Dubroca method is limited to analytical chemistry. Do you mean that, while solutes in polar organic solvents can be polarized, this is not feasible in water? Please clarify.
   **Response:** In the ODNP experiments described by Dubroca et al., the sample is directly irradiated without the use of a resonator which leads to significant sample heating. We agree that the statement about "analytical chemistry" is misleading. By not using a resonator this method will be limited to studies of samples in organic solvents (as they are often studied by NMR in analytical chemistry), that do not require precise temperature control. However, even if the sample is not destroyed by excessive heating, referencing of the chemical shifts will be challenging. The revised manuscript will be updated accordingly.
2. l. 56: dDNP experiments can be repeated, just not with the same sample. Please rephrase.
   **Response:** In a dissolution experiment the sample is diluted using a hot solvent or steam

effectively diluting the sample. To repeat the experiment, a new sample has to be prepared. 2D experiments can be performed using small pulse flip angles or gradient based acquisition methods but continuous acquisition is not possible. We will rephrase this section in the revised manuscript.

3. l. 63: The wording "sample can be pumped" is unfortunate in a hyperpolarization context. It would help to specify "sample solution can be transferred from low to high magnetic field using a pump".
   **Response:** We agree with the reviewer. This mistake will be corrected in the revised version of the manuscript.

4. Around l. 65: There is a Boltzmann factor penalty, but at the same time, enhancement is usually better at lower fields in Overhauser DNP. This should be mentioned earlier than in line 84. In general, the description of the state of the art is only loosely connected with the motivation of the present work. Please include one or two sentences that summarize the advantages and disadvantages of your approach with respect to measuring polarized samples at high field.
   **Response:** We agree with the reviewer and we will update the manuscript accordingly.

5. Standard NMR spectrometers compensate for field drifts with a field lock ("deuterium lock"). Did you consider doing this and, if so, why did you decide against it? The interleaved referencing entails a loss in measurement time by a factor of two, giving up square root of two in signal-to-noise ratio. Your effective enhancement for ethyl crotonate is only about 20 after that.
   **Response:** We use a compact, single channel NMR spectrometer, which does not have a lock channel. We deliberately decided against a traditional field lock to keep the hardware requirements as simple as possible and opted for a software solution. For the J-Res experiments a 4-step phase cycle is used and a reference spectrum was only taken after each phase cycle was completed. This results in an increase of measurement time of only 20 %. In case the field drift is small, this can be further reduced by only periodically acquiring the reference spectrum. If with "standard NMR spectrometers" a high-field NMR console is meant, we agree, the lock channel can be used to compensate for the field drift. However, many compact NMR systems for process monitoring, in-bore NMR spectroscopy, etc. do not have an extra channel that can be used as a lock, or just don't have sufficient space. We realized that we did not make this point very clear in the manuscript and we will update the manuscript accordingly.

6. l. 109: Just to be sure: "loaded resonator Q" commonly refers to the resonator being coupled to a 50-Ohm-line, not to it being "loaded" with a sample. Is it possible that 6900 is a loaded Q of the empty resonator and > 4000 is the loaded Q of the resonator with the sample inserted?
   **Response:** The reviewer is correct, the Q value of 6900 refers to the empty, but critically coupled resonator, while a value of > 4000 refers to a resonator with the sample inserted. We will correct this statement in the revised version of the manuscript.

7. l. 127: Please state explicitly that the linewidth of 0.16 ppm is not limited by shim quality but rather by the presence of the radical (you write so in l. 217). In fact, the statements in lines 216 and 217 are self-contradictory. If there is no significant line broadening due to the added paramagnetic polarizing agent, the linewidth cannot, at the same time, be

"still limited by the polarizing agent". The caption of Figure S2 indicates that the polarization agent does determine linewidth.

**Response:** We agree that this was not clearly expressed in the paper. When we state "we do not observe significant line broadening due to added paramagnetic polarizing agent" we intend to say that while the polarizing agent introduces line broadening, this does not prevent us from performing chemical shift resolved EPR. This point will be clearly expressed in the revised version of the manuscript.

The linewidth we observe for a tap water sample is indistinguishable from a sample of water with 200 μM TEMPOL. In the text we will clarify that in the case of water with 200 μM TEMPOL, the linewidth is limited by the shims.

8. Related to 7, please specify the linewidth limit of the shims. It should be possible to measure water without polarizing agent.

   **Response:** In the revised version of the manuscript we will clarify that the linewidth limited by the shims is 0.16 ppm.

9. l. 154: "This makes processing 2D NMR spectrometer data possible without requiring transposing data." Why is this an advantage? Transposing a matrix is much faster than other processing steps and the processing time for 2D data is anyway insignificant with a modern computer.

   **Response:** We agree with the reviewer that this is not an advantage with modern computers. We will instead argue the benefits of using object-oriented programming to process NMR data. The DNPLab package uses a data object which stores the axis and data in the same object. The name of each axis can be used to specify the dimension along which a processing step should be performed.

10. l. 157: Which window function was applied to 1D data? Line 214 states 1 Hz Lorentzian. What is the reason for this choice? Is this a matched filter? The numbers do not quite fit. You claim a linewidth of 2.3 Hz (l. 127, appears to be water) and a "native" linewidth of 7.3 Hz with a 1 Hz Lorentzian window (l. 214).

    **Response:** We followed standard procedures for processing liquid-state NMR spectra. The corresponding linewidth of the window function was chosen to be less than the linewidth of the ethyl crotonate to avoid significant line broadening by the window function.

11. l. 160: "a Gauss Lorentz transformation". Probably ä Lorentz Gauss transformation" is meant here, as correctly written in the caption of Figure 3.

    **Response:** This mistake will be corrected in the revised version of the manuscript.

12. If the problem with the magnet is just a slow drift that can be addressed by interleaved reference measurements, wouldn't it also be possible to correct for it by analyzing 1D FT of the individual traces of the 2D spectrum?

    **Response:** During the J-Res experiment, the signal intensity decreases with T2 and transients with longer separation times between the first and the second pulse will have significantly lower intensities. However, the signal intensity of the single-shot 1D spectrum will always have the same intensity. We therefore used the interleaved referencing method.

13. It is euphemistic to state that "some" resolution enhancement methods are known to decrease signal-to-noise ratio. Unless you know a method that doesn't entail such loss,

please delete the "some".

**Response:** We agree with the reviewer. We will rephrase this sentence accordingly.

14. l. 227: A Lorentz-Gauss transformation with the same linewidth of 4 Hz for deconvolution and convolution should not change linewidth, but only lineshape. What measure of linewidth are you using here? Please also refer to Figure S3. How did you determine the improvement from 7.3 to 5.6 Hz? What happens if you do not use Lorentzian apodization, but a Hamming window (or even Chebyshev, this is no longer problematic with current computer power).

    **Response:** We agree with the reviewer that this issue needs clarification. In our current workflow we process the data and then fit a Voigt function to the methyl proton peaks. From this we calculate the FWHM, which is given in the text. We do that because of the great overlap of the peaks. We prefer the Lorentz-Gauss transformation, because it removes the "long tail" in the JRES spectra, resulting in cleaner spectra and it removes disambiguates. For the revised version of the manuscript we will explore other apodization windows and will add the results to the SI.

[Figure]

15. l. 229 Instead of the Traficante et al. method, you could as well use a smaller convolution linewidth in the Lorentz-Gauss transformation. Did you try this?

    **Response:** Motivated by this comment, we investigated processing the data using a smaller linewidth in the Lorentz-Gauss transformation. If a window function is chosen with an equal Lorentzian and Gaussian linewidth, minimal effect on linewidth is observed. Probably the best way to use the Lorentz-Gauss transformation for resolution enhancement would be to choose a matched filter for the Lorentzian linewidth and vary the linewidth of the Gaussian until the desired resolution enhancement is achieved. We will add these results to the SI of the revised manuscript.

[Figure]

16. l. 270: "the possible applications are plentiful". Please list a few. What type of problem can be solved by high-resolution ODNP-enhanced NMR at low field (~14 MHz proton frequency) that cannot be solved by standard NMR? Your ethyl crotonate example would have better sensitivity with a 400 MHz spectrometer (your enhancement is 30). What can you potentially do that cannot be done better with standard NMR?
**Response:** A "standard NMR" spectrometer requires a superconducting magnet, cooled with liquid cryogens. While this may be the type of NMR spectroscopy that many researchers are familiar with, NMR spectroscopy is employed in many more areas that do not rely on superconducting magnet, such as process monitoring, environmental monitoring, in-bore (well) diagnostics. Many of these applications cannot be based on using superconducting magnets. Our system uses a permanent magnet, and the operating costs (mainly power for the incubator) are negligible. Any in-field method, or process monitoring application can benefit from our results.
In recent years, many low-field, NMR based diagnostic methods have emerged (e.g. see products by T2 Biosystems, WaveGuide Corp, Synex Medical to name a few), which all can benefit from ODNP enhanced NMR methods. In addition, although currently just a niche field, ODNP-based hydration studies are preferably done at a magnetic field strength of 0.35 T.

17. l. 288-289: The argument about the Boltzmann penalty is wrong. You pay the same penalty when you measure at low field. The enhancement factor is higher, but sensitivity is not higher. In fact, detection at higher field does improve sensitivity at

given polarization. The voltage induced in a coil is proportional to the rate of change of magnetic flux, which in turn is proportional to the resonance frequency.

**Response:** Hoult et al. (JMR 1976) showed that the signal-to-noise ratio scales with the power of 7/4 when comparing experiment at different field strengths, when the sample is in thermal equilibrium. When the sample is polarized at low fields prior to transferring it to higher fields (hypothetically in no time), the polarization is the same at both fields and the experiment does not benefit from the higher field. However, a) there will be a finite amount of time required to transport the sample, during which time the polarization will relax, b) by the time the system is again at thermal equilibrium, the excess polarization due to the ODNP effect has decayed, c) the ODNP effect at lower fields is typically more efficient (as mentioned by the reviewer already earlier). The overall achieved sensitivity increase is a combination of many different factors. This has been studied by Dorn et al. in the late 1980s and the achieved enhancements were always lower compared to the increase of the signal-to-noise ratio due the increased magnetic field and the ultimate conclusion was that polarization and detection should be performed at the same field for maximum sensitivity.

18. Section S3: What is the reason for the phase drift?
    **Response:** We identified several different reasons for the phase drift. First instrumental and environmental instabilities (e.g. temperature fluctuations in the room) can cause long-term phase drifts. Second, due to the acquisition delay (deadtime of the spectrometer) a frequency shift (due to the field drift) can cause a linear roll in the frequency domain. In addition, the phase of the pulse will vary with frequency as the field drifts. The interleaved referencing can take care of this effect. We will address this in more detail in the revised version of the manuscript.

19. Please refer to the ethanol data (Section S4) at some point in the main text rather than only stating that they exist.
    **Response:** We agree with the reviewer and will update the manuscript accordingly. We will refer to the ethanol data in the section about the field compensation and later when we describe the J-Res experiments.

20. Supplement l. 54/55: What kind of peak integration would properly quantify the number of protons in the ethanol spectrum?
    **Response:** To obtain quantitative information from the JRES spectrum, integration along the indirect dimension is performed. This is briefly stated in the SI, however, we will make this clear in the main text. In the manuscript, we show the skyline projection of the JRES spectrum which is a common way to present the 1D-JRES spectrum, however, the disadvantage is this is not quantitative.

Typos:
1. 10: "chemical cite" should read "chemical site"
2. 31: "present large challenge" should read "present a large challenge"
3. 140: "90-pulse length" should read "90°-pulse length"

4. 165: "were reference" should read "were referenced"
5. 176 : "which is on lower side" should read "which is on the lower side"
6. 178: "lower viscosity as for example water" should read "lower viscosity than, for example, water"
7. l. 254 "since the signal originate" should read "since the artifacts originate"

**Response:** We thank the reviewer for pointing out these typos, which will be corrected in the revised version of the manuscript.